# Preparing Cities for Future Pandemics: Unraveling the Influence of Urban and Housing Variables on COVID-19 Incidence in Santiago de Chile

**DOI:** 10.3390/healthcare11162259

**Published:** 2023-08-11

**Authors:** Katherina Kuschel, Raúl Carrasco, Byron J. Idrovo-Aguirre, Claudia Duran, Javier E. Contreras-Reyes

**Affiliations:** 1CENTRUM Católica Graduate Business School, Pontificia Universidad Católica del Perú, Lima 15073, Peru; kkuschel@pucp.edu.pe; 2Núcleo de Investigación en Data Science, Facultad de Ingeniería y Negocios, Universidad de Las Américas, Santiago 3981000, Chile; 3Escuela de Negocios, Facultad de Ingeniería y Ciencias, Universidad Adolfo Ibañez, Santiago 7941169, Chile; bidrovo@cchc.cl; 4Gerencia de Estudios y Políticas Públicas, Cámara Chilena de la Construcción, Santiago 7560860, Chile; 5Departamento de Ingeniería Industrial, Universidad Tecnológica Metropolitana, Santiago 7800002, Chile; 6Instituto de Estadística, Facultad de Ciencias, Universidad de Valparaíso, Valparaíso 2360102, Chile; jecontrr@uc.cl

**Keywords:** urban planning, housing, COVID-19, socioeconomic variables

## Abstract

In this study, we analyzed how urban, housing, and socioeconomic variables are related to COVID-19 incidence. As such, we have analyzed these variables along with demographic, education, employment, and COVID-19 data from 32 communes in Santiago de Chile between March and August of 2020, before the release of the vaccines. The results of our Principal Component Analysis (PCA) confirmed that those communes with more economic, social, organizational, and infrastructural resources were overall less affected by COVID-19. As the dimensions affecting COVID-19 are based on structural variables, this study discusses to what extent our cities can be prepared for the next pandemic. Recommendations for local decision-makers in controlling illegal immigration and investing in housing and urban parks are drawn.

## 1. Introduction

In Latin America and the Caribbean, the SARS-CoV-2 (COVID-19) pandemic exacerbated public health challenges and opened the debate on security when facing biological threats and national supply. The COVID-19 pandemic affected the population in a different way depending on the socioeconomic level of each segment [1], strategic geographical location—especially in cities with airports or high-speed trains [2,3,4], high density, and with subnormal agglomerations such as favelas, caves, lowlands, irregular subdivisions, huts, and houses on stilts [5,6]. Depending on the different types of housing, the available physical space, and social distance, the COVID-19 propagation and infection rate varied widely [7,8]. The health emergency caused social and economic damage, such that it was necessary to generate more effective and efficient public policies that could help the most vulnerable sectors [9], since the pandemic created new inequalities in several aspects [10]. To face this situation and to take effective measures to help people, the need to design new public policies in the social, economic, and public health dimensions emerged [11,12].

The actions of the governments were limited by public budgets that quickly had to prioritize the distribution of resources and by bureaucracy and technology constraints, which depended on the level of technical efficiency of each country [13]. Aspects of public policies that became priorities were made visible, which related to the economic situation, culture, health and the degree of technology that each country had [9,13,14]. Along with the pandemic, difficulties arose in decision-making for the governments, wherein public health and management of patients and mortality were the most challenging issues [9]. At the beginning of the pandemic, as of 31 June 2020, there were 508,055 deaths worldwide, and in Chile, between March and June 2020, 10,105 deaths from COVID-19 were registered [15].

The complexity in Latin American countries was magnified by the lack of information and technical knowledge regarding epidemiology and public health crisis management, which affected public administration. Scarce hospital resources had to be allocated, and challenges on multiple fronts required attention, such as education, poverty and unemployment, access to health and health insurance, mental health, and physical well-being [9,16]. These difficulties were greatest for the most vulnerable population such as the disabled, immigrants, the unemployed, and the homeless people [17].

On the other hand, the social isolation policies that had been applied impacted the country’s macroeconomic indicators [18], decreasing the Gross Domestic Product (GDP) per capita and increasing public spending [19]. According to the Organization for Economic Cooperation and Development (OECD), since the beginning of the COVID-19 pandemic, Chile was among the three Latin American countries with the highest health expenditures per person, increasing from 9.65% in 2019 to 12.41% in 2020, and it is expected that in 2050 health spending per person will rise to 14% of the GDP [20,21].

Due to the complexity of the situation, since the beginning of the pandemic, different mathematical models were created in order to determine the causes, effects, and social and economic trends that affected the population to prevent the risk of COVID-19 infection [22]. In these models, relevant variables were identified such as the socioeconomic aspects captured by the Gini Index and GDP and the air quality variables [23], geo-climate, and demography, which helped to explain why the effects of contagion differed from one region to another [24,25].

### Motivation

In Chile during the pandemic period, different public and private institutions collected data and performed studies to determine the key aspects of the impact of COVID-19 in the communes of the capital city Santiago. Moreover, real-time information on the behavior of contagion cases and mortality was made available to the public [26]. However, COVID-19 generated in this country—as well as in others—a systemic problem in which huge deficiencies were detected in health care and economic security [26] that revealed the need to further study the relationship between socioeconomic and geo-demographic variables regarding the pandemic management.

## 2. Literature Review

This section addresses the associations between the dynamics of registered COVID-19 cases and sociodemographic situation, urban planning, and housing policies [27] in a global context, along with a brief description of the context of Santiago de Chile.

### 2.1. Socioeconomic Status, Lifestyle, and Education

The relationship between socioeconomic variables and the different measures of COVID-19 (i.e., affected, confirmed, active, and deceased cases) was extensively studied at the beginning of the pandemic in order to identify which other groups of people could be vulnerable, and try to predict the rate of infection and mortality.

Data were also analyzed at the country level, taking into consideration infrastructure, number of physicians, and health expenditure. Regarding the GDP, higher GDP per capita was related with fewer cases of COVID-19 [28].

The proportion of people over 65 years of age is also positively related to a higher number of infected cases [29]. Regarding death rate, South American countries with higher population density had paradoxically lower numbers of deaths [30]. Moreover, the study conducted by Canatay et al. [31] showed that power distance, individualism, gender, and age affected the death rate more than other socioeconomic factors explored. COVID-19 patients with low educational level in India had significantly more complications and higher mortality [32].

### 2.2. Urban Planning

Pollution and population size were also associated with COVID-19 cases. Improvements in air and water quality in cities during lockdown periods emphasized the environmental consequences of human activities [33]. High population density was one of the most important factors for the quick spread of COVID-19, as social distance was difficult to maintain [34]. However, the spread of COVID-19 was not linked with population density in the USA [35].

### 2.3. Housing

The COVID-19 pandemic rapidly emerged as a housing emergency [36]. A study conducted in Washington DC, US, found that housing quality and living conditions were strongly correlated with COVID-19 death count [37]. On the other hand, Latin American countries faced extra issues. Inhabitants of slums were especially impacted by the pandemic, not only because of the physical impossibility of social distancing due to high density [5], poor-quality housing, or the lack of water to wash their hands [6,38] but also due to other specific issues in Low and Middle Income Countries (LMIC). Residents of LMIC had increased risks for respiratory infections due to elevated levels of air pollution [39], and during the social isolation phase, the expansion of the rodent population aggravated the risk of zoonotic diseases [40]. Moreover, connectivity played a central role in the spread of COVID-19 [41,42]. However, as found by Gargiulo et al. [43], significant correlations were not always identifiable between settlement characteristics and the spread of COVID-19. According to them, the diffusion of the new coronavirus was closely related to main features of the demographic and socioeconomic structure of the urban population.

### 2.4. Antecedents: Chile and COVID-19

Chile was recognized worldwide as one of the Latin American countries that during December 2020 to June 2020 made remarkable progress in the rate of inoculation against COVID-19 through vaccination; this was demonstrated in the results obtained from the percentage of vaccinated people over the size of the population (Source: https://ourworldindata.org/coronavirus (accessed on 30 june 2022)). Globally, in the period from December 2020 to April 2021, it was ranked third among countries whose populations had received at least one dose, and second for all doses according to the vaccination protocol [44]. Figure 1 shows that the commune of Independencia had the highest index of Number of Confirmed Cases (NCOC)/population index, due to the fact that this commune is the main recipient of immigrants, which causes high levels of overcrowding.

Government authorities, with the support of private sector institutions, developed the health plan known as step by step [45]. This plan standardized the measures (non-pharmacological norms) that were applied to the inhabitants clearly establishing the conditions of exit or return from confinement for each commune of Chile, according to their level of improvement or deterioration of their health status reflected in the results of the COVID-19 indicators. In this way, the restrictions on mobility were dynamically differentiated since they did not have the same severity and duration throughout the country [45].

It is worth mentioning that during the beginning of the pandemic in the days of total quarantine when the city was completely closed, efforts were made in the city of Santiago to identify and measure both the level of poverty and the mobility of people. This was done with data obtained from the mobile phones within each commune and the comparison with the average population living in poverty [46].

In relation to Socioeconomic Status (SES), between March and June 2020, the 32 communes of Santiago and two neighboring municipalities were measured, and it was obtained as a result that the two municipalities with the highest SES exhibited a reduction in mobility by up to 61 percent, while this rate was only 40 percent among the municipalities with low SES [26]. It was observed that the inhabitants of the poorest areas of Santiago had lower incomes and experienced a greater economic impact due to the quarantine measures [26,47].

## 3. Materials and Methods

### 3.1. Materials

A sample was selected with 32 records corresponding to the communes of the Province of Santiago.

The variables as described in this article were created from the data of the following sources: the National Socioeconomic Characterization Survey (CASEN (Source: http://observatorio.ministeriodesarrollosocial.gob.cl/encuesta-casen (accessed on 16 February 2023).)), the national system for the evaluation of learning results (SIMCE(Source: https://informacionestadistica.agenciaeducacion.cl/#/bases (accessed on 16 February 2023).)), the urban quality of life, the social priority index, among other variables.

### 3.2. Variable Selection Using GLS-Based Linear Regression

To identify the causal relationship of the urban environment quality variables and the housing deficit in the performance of health indicators, the Generalized Least Squares (GLS) estimation method [48] was applied to correct in a simple and intuitive way the problem of the correlated residuals, given the high probability of finding heteroscedastic residuals mainly due to the analysis of cross-sectional data. Therefore, the estimation by GLS was considered instead of the estimates obtained by Ordinary Least Squares (OLS) due to the possible presence of heteroscedasticity and/or autocorrelation.

Given that socioeconomic data and health indicators distributed in 32 communes of Santiago were studied, the residuals were expected to reflect a pattern of greater dispersion (or heteroscedasticity) around the mean of the observations; however, this measure of central tendency tends to concentrate mixtures of data, making it difficult to correctly identify the relationship between the variables of interest. For example, regarding socioeconomic variables such as the deficit caused by involuntary allegation, the high or low deterioration level could reflect a greater or lower risk of contagion of the COVID-19 virus, respectively, and thus a greater or minor impact on health indicators. However, around the average value there is an overlap with the socioeconomic situation of households, so the dispersion of its relationship with the performance of health indicators is greater. In this context, the application of the GLS method can obtain robust estimates of the coefficients that accompany the predetermined variables of the regression models.

Formally, consider the following set of linear regression models M=(j×i) with heteroscedastic structure on the residuals:(1)$$y_{i}^{\left(j\right)}=x_{i}^{\left(k\right)}\beta^{\left(j,k\right)}+\varepsilon_{i}^{\left(j,k\right)},\;\;\;\;\;\;\varepsilon_{i}^{\left(j,k\right)}∼N\left(0,{\sigma_{i}^{2}}^{\left(j,k\right)}\right)$$
where $y_{i}^{\left(j\right)}$ is the *j*-th health indicator of the commune *i*; {j,i}∈N×N; $x_{i}^{\left(k\right)}$ is the *k*-th socioeconomic variable that measures the situation of the urban environment or characteristic of the household located in the commune *i*; $\beta^{\left(j,k\right)}$ is the coefficient of the *M*-th regression model that measures the impact of the *k*-th socioeconomic variable on the *j*-th health indicator (or dependent variable); $\varepsilon_{i}^{\left(j,k\right)}$ is the residual associated with each regression model, which is assumed to follow a normal distribution with zero mean and heteroscedastic variance $\left({\sigma_{i}^{2}}^{\left(j,k\right)}\right)$. The assumption of this initial structure of the linear regression model is made since the presence of heteroscedasticity is common in cross-sectional observations. It was included in the process of estimating the parameters of regression models, since, on the contrary, these would be far from optimal in light of the perspective of the Gauss–Markov theorem [49].

The GLS method consists of pre-multiplying all observable variables of the regression model by the inverse of the square root of the variance of the error term:(2)$$\frac{y_{i}^{\left(j\right)}}{\sqrt{{\sigma_{i}^{2}}^{\left(j,k\right)}}}=\frac{x_{i}^{\left(k\right)}}{\sqrt{{\sigma_{i}^{2}}^{\left(j,k\right)}}}\beta^{\left(j,k\right)}+\mu_{i}^{\left(j,k\right)},$$
where, according to the construction, it is defined by
(3)$$\mu_{i}^{\left(j,k\right)}=\;\frac{\varepsilon_{i}^{\left(j,k\right)}}{\sqrt{{\sigma_{i}^{2}}^{\left(j,k\right)}}}∼N\left(0,1\right).$$

In this sense, the variance of the error term changes from a variable to a constant throughout all the observations of the sample of communes of Santiago. So, ${\hat{\beta }}_{GLS}^{\left(j,k\right)}$ is a consistent estimator of $\beta^{\left(j,k\right)}$, i.e., asymptotically unbiased and efficient. In this regard, it is worth mentioning that this procedure is also known as White’s estimator.

As the estimation strategy, the GLS method was applied in two steps:(i)Initially, an approximation of the parameter ${\sigma_{i}^{2}}^{\left(j,k\right)}$ is made from the OLS estimator of the term ${\hat{\varepsilon }}_{i}^{\left(j,k\right)}$ by raising each observation to the second power. So, ${\hat{\varepsilon }}_{i}^{2(j,k)}\approx \;{\hat{\sigma }}_{i}^{2(j,k)}$;(ii)The inverse of the square root of the estimator obtained in (i) is multiplied by each observed variable of the regression model, so as to re-estimate the model transformed by the OLS method.

The estimator of $\beta^{\left(j,k\right)}$ resulting from steps (i) and (ii) is defined as the estimator of GLS $\left({\hat{\beta }}_{GLS}^{\left(j,k\right)}\right)$. This method was applied only in cases wherein the homoscedasticity hypothesis was statistically rejected.

### 3.3. Principal Component Analysis

After having selected the set of variables by linear regression based on GLS (Section 3.2), the Principal Component Analysis (PCA) method was used in order to classify them into groups or dimensions of variables. PCA is a multivariate statistical technique used to reduce the dimensionality of a dataset by identifying underlying patterns in the data [50]. It seeks to transform the original variables into a set of uncorrelated variables called “principal components”, which are ordered according to the amount of variance they explain in the data [50].

In this sense, these principal components or dimensions summarize the information of the set of socioeconomic variables, thus making it possible to characterize the infected sectors in Santiago and to identify risk areas [51,52]. This approach has the advantage of directly analyzing the hot spots of infection in Santiago de Chile over other linear methods that could provide unreliable and noisy data [53].

### 3.4. Implementation of the Methods

For the selection of dependent and independent variables that can be used as an instrument, the following steps and criteria were taken:1.Selection of variables using linear regression based on GLS:(a)A correlation coefficient greater than 60% ($R^{2}$ adjusted > 0.60) is considered;(b)They are classified as exogenous variables if they are not statistically correlated with the residuals and do not present bias and endogeneity. Then:i.Two groups are obtained that are divided into those with correlation 0, and those whose correlation is significant;ii.Regressions correlated with the residual are discarded or, if necessary, corrected using GLS.(c)White’s heteroscedasticity test is applied to all variables to verify the contribution of each of the variables to variance, using the hypothesis test:i.$H_{0}:$ there is no heteroscedasticity;ii.$H_{1}:$ there is heteroscedasticity, because the variance is not constant.(d)The Bayesian information criterion (BIC) is used to select the best model (lowest BIC) from a finite set of possible *M* models, which can define the cause and effect variables applicable to the pandemic period.2.Using PCA, a multivariate analysis was performed on the COVID-19 dataset. In addition, the dependence of the influential variables was studied, the correlations were identified, and clusters were determined by *k*-means (For the PCA and *k*-means, the library factoextra [27], implemented in the R software (R version 4.2.3, https://www.r-project.org/ (accessed on 30 March 2023)), was considered).3.Finally, a broader socioeconomic index was constructed, which included the independent variables related to the following aspects: income, education [54], occupation [55], and house structure [48].

## 4. Results

The results were obtained by applying the steps and criteria as shown in Section 3.4 to the data that were collected during the COVID-19 pandemic period during March and August of 2020. The variables were identified and classified according to their characteristics as follows:1.Dependent variables: Number of Confirmed Cases (NCOC), Cumulative Incidence Rate (CMIR), Number of Deceased (ND), Mortality Rate (MR), Number of Current Cases (NCUC), Current Incidence Rate (CRIR), Number of Active Cases (NAC), Asset Incidence Rate (AIR).2.Independent variables: Electricity Consumption Per Capita Residential (ECPCR), Private Homes that Require Materiality Improvements (PHRMI), New Urban Housing Requirement (NUHR), Percentage of Overcrowded Dwellings (POD), Percentage of Dwellings with External Allegiance (PDEA), Park User Population (PUP), Number of households that live in the same house, with an independent budget (HSH), Labor Conditions (LC), Housing and Environment (HE), Social Priority Index (SPI), Social Register of Households (SRH), Average Income of Unemployment Insurance Affiliates (AIUIA), System for Measuring the Quality of Education, Reading (EQMSR), Education Quality Measurement System, Mathematics (EQMSM), University Selection Test (UST), Potential Life Years Lost (PLYL).

Figure 2a shows the main results of the PCA, in particular the percentage of variance explained by the variables related to COVID-19 for each of the dimensions. It can be seen that dimension 1 and 2 concentrate the greatest variance of the system with 50.4% and 37.9%, respectively. On the other hand, Figure 2b shows the percentage of the variance explained by the independent variables for each of the dimensions. It is noted that, since dimension 1 represents the largest proportion of the system with 75.3%, the others are discarded. Dimensions with largest proportions correspond to the most significant factors in this study and are broken down in Figure 3 as follows, where analyzed variables are grouped together for each of those factors.

From the results of the PCA, the contribution of the variables to each dimension can be noted in Figure 3a,b; in the diagrams, the correlations between the first five dimensions are shown with the variables related to COVID-19 and the independent or explanatory variables, respectively. It was obtained that the highest correlations were determined by the first two dimensions related to COVID-19 (see Figure 2a), and by the first dimension related to the independent variables (see Figure 2b).

Figure 4a,b show the results of the application of the K-means technique in the socioeconomic dimension and COVID-19. They are grouped by sectors of the Province of Santiago:Cluster 1: corresponds to the southeast, southwest, northwest, south, and central sectors which have a medium–low level of socioeconomic development and include the following communes: Macul, La Florida, Maipu, Cerrillos, Santiago, Huechuraba, Pudahuel, Quilicura, San Miguel, and La Cisterna.Cluster 2: incorporates the northwest, north, and south sectors that include the communes with the lowest level of socioeconomic development such as Cerro Navia, Lo Prado, Conchali, Independencia, Recoleta, El Bosque, Pedro Aguirre Cerda, San Joaquín, Lo Espejo, La Granja, San Ramon, La Pintana, Quinta Normal, and Renca.Cluster 3: located in the northeast, it contains the communes of greater socioeconomic development such as La Reina, Las Condes, Lo Barnechea, Ñuñoa, Providencia, and Vitacura.In the case of the communes of Cerrillos, Estacion Central, and Peñalolen, they are grouped into different clusters depending on the COVID-19 dimension (see Table 1).

## 5. Discussion

This section discusses the results obtained on the influence of (1) population density and (2) housing and urban parks on COVID-19 disease in Santiago de Chile, noting that national and local governments would have little power in the short term to intervene and stop a pandemic by modifying the variables found by our study.

The PCA results (see Figure 3b) showed that most of the explained variables in dimension 1 called “economic variables” can be categorized into three groups: sociodemographics (e.g., electricity consumption per capita, social priority index), housing situation (e.g., homes that require improvements, new urban housing requirement, overcrowded dwellings, dwellings with external allegiance, park user population, households in the same house, housing and environment, social registry of household), and education and employment level (e.g., labor conditions, income of unemployment insurance affiliates, reading skills, mathematical skills, university selection test). Then, the communes of Santiago were organized into three clusters according to the dimensions (see Figure 4). There is almost a match between the explained COVID-19 variables and the list of variables detailed above. In other words, the analysis confirmed that those communes with more economic, social, organizational, and infrastructural resources were overall less affected by the COVID-19 pandemic.

Moreover, the education test scores also appeared as variables that cannot be quickly changed from one year to another. Even worse, as illegal immigrants mainly arrive to slums, education policies related to citizenship need to be strengthened as a way to build social cohesion [56] and, thus, to avoid an impact on scholar performance, crime increase, and the building of ghettos that threaten national security [57].

Population density has been found to be a key player in the spread of respiratory viruses. According to the last Chilean census in 2017, the population of Santiago was approximately 5,250,565 people, equivalent to about 28.4% of the national population. Said value was expected to increase to 6,075,403 by 2021 [58] for the complete capital (32 communes). Since the past decade, Santiago has been receiving many immigrants, mainly from Venezuela and Haiti [59]. In total, the Metropolitan Region has the largest number of foreigners with 909,414 people. According to official data, the Metropolitan Region received an influx of 96,464 in the 2018–2021 period [60].

It is known that illegal immigrants are not able to work or rent a house. As such, the influx of immigrants directly impacts the variables that explained the COVID-19 pandemic, such as housing requirements, overcrowded dwellings, and labor conditions. However, as reflected in our results, municipalities can use urban parks as a tool with the potential to ameliorate the virus spread and improve people’s mental and physical health [61] as a short term solution. In spite of this, a more stable solution to COVID-19 spread, severity, and mortality is the investment in housing facilities and the control of informal immigration, which, on the contrary, will increase population density and stress the variables in the dimension.

The effectiveness of pandemic mitigation strategies depends not only on the willingness of the population to comply with them but also on their true capacity to do so [62]. Once the emergency is over, slum communities will have to cope with social and economic reactivation and react effectively to the multiple social and psychological consequences, new waves of COVID-19 infection, or other pandemics [38].

Finally, the population’s access to quality health insurance was restricted to the socioeconomic factor, since people of low socioeconomic status did not reduce their mobility during confinement as much as those living in more prosperous communes [26]. Moreover, mortality rates in youth of low socioeconomic status were higher in low-income municipalities. In addition, there was a strong impact of the confinement measures (restriction on freedom of movement, dynamic quarantines, and suspension of non-essential activities) carried out by the Chilean government on the working and economic conditions of the Chilean citizens and even more dramatically on immigrant population [59].

## 6. Conclusions

This study addressed the research question of how the COVID-19 pandemic variables are correlated to urban, housing, and socioeconomic variables. Urban, housing, socioeconomic, demographic, education, employment, and COVID-19 pandemic-related data were collected from 32 communes of Santiago before the release of the vaccines. The PCA confirmed that those communes with more economic, social, organizational, and infrastructural [26] resources were overall less affected by COVID-19 disease.

### 6.1. Implications for the Preparedness for Future Pandemics

Our results serve as evidence for local decision-makers in making changes, such as controlling illegal immigration and investing in housing and urban parks. The structural nature of the variables used in this study, show that if urban, housing, and socioeconomic characteristics stay constant, the municipalities could face incidence rates in a future respiratory disease pandemic similar to those during the early stages of COVID-19.

Identifying the most vulnerable people in a pandemic could improve logistical capacities, enhance health systems, protect employment, support entrepreneurship, and prioritize the social protection provided to people in the most at-risk communities [63,64]. Implementing new forecasting methods has some disadvantages, such as the need for quality data, which requires more technical knowledge of operational and strategic data analysis, and may increase the technological costs of implementing integrated online monitoring systems [65].

On the other hand, the health crisis caused by the pandemic forced the state to take measures without any prior planning or diagnosis to identify the critical aspects to address [66]. The lack of data and historical information available in 2020 emphasized the need to review and create new public policies that could be useful to improve political decision-making and support both families and small and medium-sized companies [67,68]. For example, there was a need to coordinate private and public actors, to follow the traceability of epidemiological and virological data, to establish new protocols, to develop surveillance strategies, and to establish control mechanisms [69].

### 6.2. Future Research Questions

Our study took cross-sectional data until the second wave of COVID-19 pandemic in 2020. The vaccines were administered free of charge to the population from December 2020 onward. However, as in many other countries, some people were reluctant to get the vaccine. We are interested to know how our model changes if vaccine coverage is included.

Analysis was carried out with available data of 2020, the critical period of the pandemic in Chile. It is expected to consolidate the results by considering an updated dataset that includes 2021. With a multivariate prediction in a future pandemic, an early and more complete diagnosis could be made in a city for which the most affected people and the relationship between the disease and the most vulnerable sectors can be determined, and thus public policies can be created that help the state to make more automated decisions in real time in a segregated city.

## Figures and Tables

**Figure 1 healthcare-11-02259-f001:**
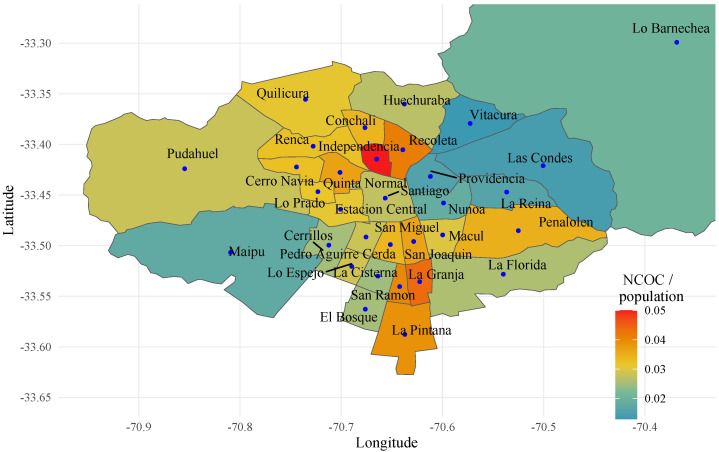
Number of Confirmed Cases (NCOC)/population index, to 18 June 2020, https://www.minciencia.gob.cl/covid19/ (accessed on 29 September 2020).

**Figure 2 healthcare-11-02259-f002:**
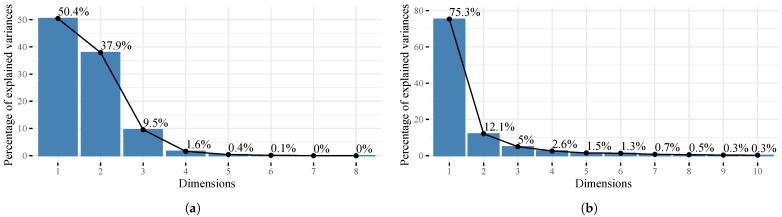
Resulting dimensions from the PCA for both (**a**) COVID-19 and (**b**) explanatory variables.

**Figure 3 healthcare-11-02259-f003:**
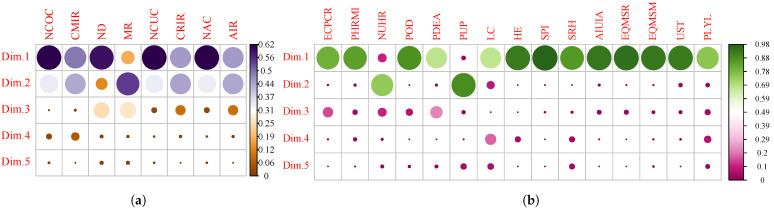
Variable contribution to each dimension. Circle’s size represents the correlaton level. (**a**) COVID-19, Acronyms: Number of Confirmed Cases (NCOC), Cumulative Incidence Rate (CMIR), Number of Deceased (ND), Mortality Rate (MR), Number of Current Cases (NCUC), Current Incidence Rate (CRIR), Number of Active Cases (NAC), Asset Incidence Rate (AIR) and (**b**) Variables, Acronyms: Electricity Consumption Per Capita Residential (ECPCR), Private Homes that Require Materiality Improvements (PHRMI), New Urban Housing Requirement (NUHR), Percentage of Overcrowded Dwellings (POD), Percentage of Dwellings with External Allegiance (PDEA), Park User Population (PUP), Number of households that live in the same house, with an independent budget (HSH), Labor Conditions (LC), Housing and Environment (HE), Social Priority Index (SPI), Social Register of Households (SRH), Average Income of Unemployment Insurance Affiliates (AIUIA), System for Measuring the Quality of Education, Reading (EQMSR), Education Quality Measurement System, Mathematics (EQMSM), University Selection Test (UST), Potential Life Years Lost (PLYL).

**Figure 4 healthcare-11-02259-f004:**
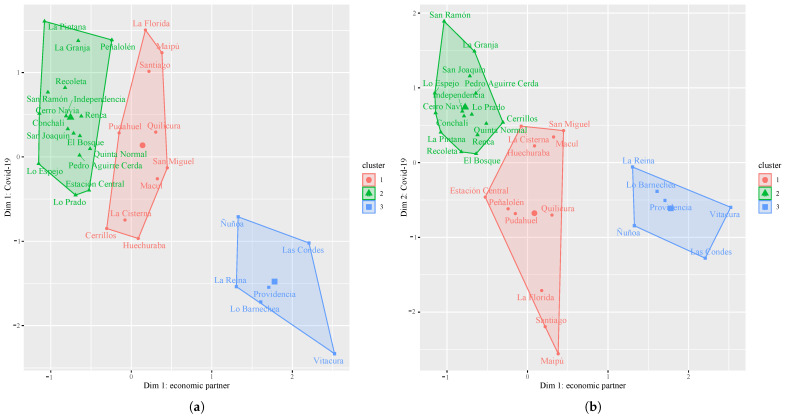
Clusterization via K-means: (**a**) Dim 1: COVID-19 and (**b**) Dim 2: COVID-19.

**Table 1 healthcare-11-02259-t001:** Dimension 1 and 2 for communes Cerrilos, Estación Central, and Peñalolen.

Commune	Dim 1: COVID-19; Cluster	Dim 2: COVID-19; Cluster
Cerrillos	1	2
Estación Central	2	1
Peñalolen	2	1

## Data Availability

Not applicable.

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
