# Peer review of "Preparing Cities for Future Pandemics: Unraveling the Influence of Urban and Housing Variables on COVID-19 Incidence in Santiago de Chile"

_healthcare, 2023, doi:10.3390/healthcare11162259_

Round 1

Reviewer 1 Report

The topic of this paper is very much in line with the future urban development, especially the considerations applied to the prevention and control of epidemic transmission, and through the exploration of city-related variable characteristics such as urban, housing and socio-economic changes. Although the consideration and application of research methods are scientific and logical, the relationship between social economy and urban space and residents' attributes is very complex. Therefore, it is relatively weak to seek supporting evidence and discussion by using only linear research methods. However, when it comes to the concern that urban residents are exposed to possible epidemics, the research results can provide government the way of consideration and control of several key impact factors as described in the study. As a consequence, I suggest that the limitations and possible inadequacies of this study can be further explained in the conclusion. In addition, it can provide policy guidance and ideas for emergency management and urban socio-economic development when considering extendable research topics and public health issues for future cities.

Reviewer 2 Report

This study addressed the research question of how the COVID-19 pandemic variables are correlated to urban, housing, and socio-economic variables. Principal Component Analysis (PCA) confirmed that those communes with more economic, social, organizational, and infrastructural  resources were of the overall less affected by COVID-19 disease.

By using the suggestions of the authors for future pandemic, an early and more complete diagnosis could be made in a city for which the most affected people and the relationship between the disease and the most vulnerable sectors can be determined; and thus, public policies can be created .

Reviewer 3 Report

This article analyzes this relationship between the COVID-19 pandemic variables and the urban, housing, and socioeconomic Chilean variables. The overall context of the article is clear and lucid It is very easy to understand, but there are a few issues to be aware of:

1. the abstract is confusing. Suggest reorganization.

2. Suggest adding the strengths and weaknesses of this study.

3. References are not formatted consistently, e.g., 13, 17, 20. Please review and standardize the reference format.

Reviewer 4 Report

The research question could be clearer and seems to be described differently in the abstract than in the conclusion. The abstract refers to 'variables beyond the common epidemilogical or biostatistical approach' wheras the conclusion refers to 'urban, housing nd socioeconomic variables' which are often used in epidemiological studies.

The dimesnions (or components or variables), referred to in Figure 2, are not adequately explained.  Given the complexity of the statistical analysis, this makes it very difficult for the average reader to distinquish which variabke contributed most to the variations in spread of Covid-19 or morbidity or mortality rates.

The key finding appars to be that 'economic, social, organisational and infrastructural resources' reduced the burden of Covid-19.  Although This is an important if unsurprising finding but begs the key question of what sort of interventions might address this problem before the next pandemic.

I felt that Figure 1 (which describes the distribution of cases within sub-areas of Santiago) is not clearly explained/discussed in the text; for example, that these are numbers rather than rates, don't necessarily reflect mortality.  There is no acknowledgement that this may not accurately reflect true incidence because more deprived areas may have more under-reporting of cases.

This is a paper with complex analysis. I feel the .presentation of results could be clearer.

On a minor point, I noted the phrase 'power distancing' on line 82 which I assume means communities or individuals more distant from those with the power of decision making; a small explanation in parenthesis would have helped.
